# Quasi-Independent Bidirectional Communication Methods for Simultaneous Wireless Power and Information Transmission

**Rongxiang Yuan, Pilong Guo \***, **Changsong Cai and Lin Yang**

School of Electrical Engineering and Automation, Wuhan University, Wuhan 430072, China;
rxyuan@whu.edu.cn (R.Y.); changsongcai@whu.edu.cn (C.C.); yang_lin@whu.edu.cn (L.Y.)

\* Correspondence: pilongguo@whu.edu.cn; Tel.: +86-186-7174-2745

**Abstract:** Simultaneous wireless power and information transmission (SWPIT) is essential for power regulation, load detection, and maximum efficiency tracking in wireless power transfer (WPT) systems. To eliminate the interaction between power and information transmission, this paper proposes a novel bidirectional communication method based on differential phase shift keying (DPSK). First, the steady-state and transient models of the series–series compensated WPT system are established to analyze the characteristics of power transmission and the behavior of shared power/information channel. Second, regarding the stability of power transmission, principles of the forward and backward communication that integrated with power modulation through the full-bridge inverter and the semi-bridgeless active rectifier are elaborated. Then, the demodulator and half-duplex communication protocol are designed. Finally, the simulation is carried out and the experimental prototype is established, which shows that the proposed DPSK method is effective and the transmission of power and information is quasi-independent. Owing to the integration of power and information transmission, the complexity, cost, and volume of the SWPIT system are reduced significantly.

**Keywords:** wireless power transfer; simultaneous wireless power and information transmission; bidirectional communication; differential phase shift keying

## 1. Introduction

Wireless power transfer (WPT) technology can provide energy without physical contact. Due to its convenience, safety and waterproof characteristics, this technology has been widely used in many fields, such as mobile phones [1], electric vehicles [2,3], smart grids [4,5], and medical devices [6]. In most applications, the load resistance and coupling inductance of the WPT systems are uncertain; therefore, a real-time information exchange is essential for primary and/or secondary side controller to maintain the stability of power transmission.

Recently, simultaneous wireless power and information transmission (SWPIT) technology has attracted considerable attentions due to its high integration feature. According to the direction of information transmission, it can be divided into forward, backward and bidirectional communication. For forward communication, the information of primary side is sent to adjust equivalent impedance [7] and track maximum efficiency [8] by the secondary side controller. For backward communication, the information of secondary side is sent to regulate power capacity [9] and realize zero phase angle (ZPA) [10] by the primary side controller. Compared with single-side communication, bidirectional communication can exchange more information for controllers, and it is easy to realize intelligent wireless power transmission [11].

For the implementations of bidirectional communication, there are four basic methods, as shown in Figure 1, where "TX", "RX" and "COM" represent the primary power converter, the secondary power converter and the compensation circuit, respectively. For the first method, as shown in Figure 1a, radio frequency (RF) link technologies, such as WiFi, Zigbee and Bluetooth, are used to exchange information between two sides. Since the power module and communication module are designed independently, the interaction between power and information transmission can be ignored. However, the RF link is subject to EMC problems [12] and may not work properly, especially in the high-power applications.

For the second method, a specially designed communication coil placed around power coil is used to exchange information between the two sides, as shown in Figure 1b. To meet strict space constraints, the communication coil can be arranged on coaxial plane [13,14] or orthogonal plane [15]. Since the frequency of data carrier is not limited by the frequency of power carrier, this method can achieve high data rate. Unfortunately, the data signal is easily affected by the power loop, resulting in low signal-to-noise ratio (SNR) [16]. Moreover, the volume and cost of WPT system are increased significantly.

For the third method, information transmission is realized by coupling with compensation circuit, where the data signal is injected from one side and extracted from other side [17], as shown in Figure 1c. The data channel and power channel are shared in this method, and it is promising to realize high integrated SWPIT system. However, the data signal is much smaller than the power signal, and the transmitted data may not be detected from demodulator [18,19]. To improve the data signal quality, the power level of communication system should be increased. Thus, the power consumption of communication circuit can not be neglected in this method, especially in the low power scenarios [20].

Compared with the previous three methods, the fourth method based on modulating with power converter, as shown in Figure 1d, has minimal hardware consumption and is compatible with a wide range of applications [21–25]. In [22], a full-duplex bidirectional communication is achieved by adjusting duty cycle of the flyback converter and calculating the times of resonances. In [24], the operating frequency and duty cycle of power converters on both sides vary with the transmitted information. However, there are still two problems in the fourth method: (1) Since the communication process causes the WPT system to deviate from its normal operating statues, the output power may fluctuate, and the power transmission performance is affected by the information transmission [26,27]; (2) The data-frequency-ratio is limited by the non-differential modulation method, and the operating frequency of power carrier can not be fully utilized.

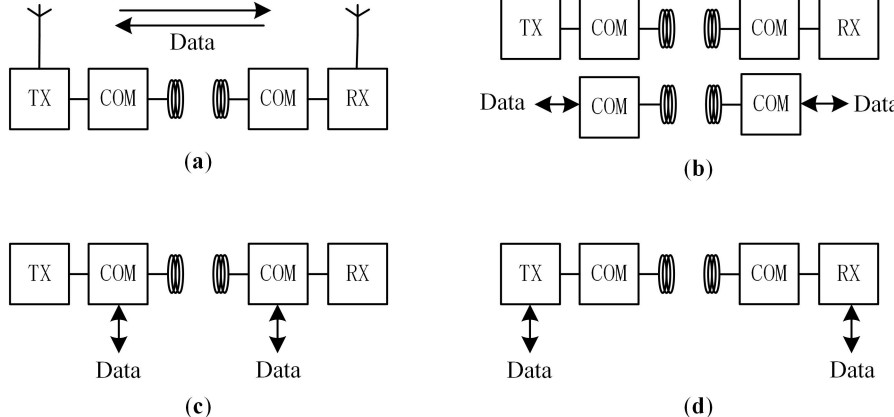

**Figure 1.** Diagram of four basic methods for bidirectional communication. (**a**) Radio frequency link. (**b**) Additional communication coils. (**c**) Coupling with compensation circuit. (**d**) Modulating with power converter.

To address these problems, this paper proposes a novel differential phase shift keying (DPSK) modulation method for simultaneous wireless power and information transmission. The main advantages of proposed method are summarized as follows:

- By using the DPSK modulation method, the power transmission is quasi-independent of information transmission and remains stable during communication process.
- Compared with the non-differential modulation, the operating frequency of power carrier is fully utilized, the date-frequency-ratio is increased.
- Not only the inductive link but also the power/data modulator is shared, which greatly reduces the complexity, cost and volume of the SWPIT system.

The remainder of this paper is organized as follows. In Section 2, the proposed DPSK scheme is briefly introduced, and the steady-state and transient characteristics of the capacitor-compensated SWPIT system are analyzed. In Section 3, the communication system is described in detail, which includes modulator integrated with power converter, demodulators and half-duplex communication protocol. The simulation and experiment are performed in Section 4. Finally, the conclusion is drawn in Section 5.

## 2. System Overview and Circuit Analysis

### 2.1. System Overview

Figure 2 depicts the schematic diagram of the proposed SWPIT system based on the DPSK method. The system can be divided into three units: power supply unit, forward communication unit and backward communication unit. For the power supply unit, a full-bridge inverter is used to convert the DC source into the high-frequency AC source. To eliminate reactive power caused by the low-coupling coils, two compensation capacitors ($C_p$ and $C_s$) are inserted between the power converters and the coils. On the secondary side, conventional passive rectifier is replaced by a semi-bridgeless active rectifier (S-BAR) to regulate the equivalent resistance and output power. For the communication unit, the transmitted data (Data in) are respectively sent to modulators for regulating phase-shifting angles ($\varphi_p$ and $\varphi_s$) and driving the full-bridge inverter and the S-BAR. Then, by detecting the instantaneous current ($i_p$ and $i_s$) in the compensation circuits, the received data (Data out) is recovered from demodulator. Notably, the modulator not only modulates transmitted data, but also regulates the voltage on the primary side or the impedance on the secondary side. Owing to the modulator and inductive link are shared, the cost and volume of the designed SWPIT system are significantly reduced.

### 2.2. Power Transmission Characteristics of SS Compensated Circuit

As mentioned in the previous Section, the information and power channels share the same modulator, so it is necessary to evaluate the power transmission characteristics. In order to simplify the analysis, a common serial–serial (SS) compensation circuit is selected and studied. As shown in Figure 3, the equivalent circuit based on the mutual inductance model is established.

The operating frequency studied in this paper covers the Qi standard and SAE J2954 recommended practice, which ranges from 80 kHz to 200 kHz. Ignoring the stray capacitance of coils and semiconductors and supposing the WPT system in the resonant state [28], the operating angular frequency $\omega$ of the system is given by

$$\omega = \frac{1}{\sqrt{L_p C_p}} = \frac{1}{\sqrt{L_s C_s}} \tag{1}$$

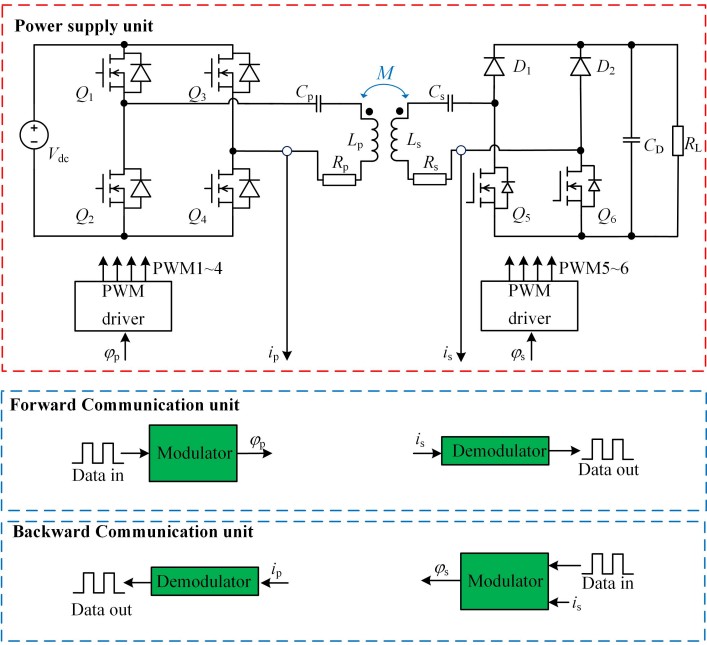

**Figure 2.** Schematic diagram of the proposed simultaneous wireless power and information transmission (SWPIT) system.

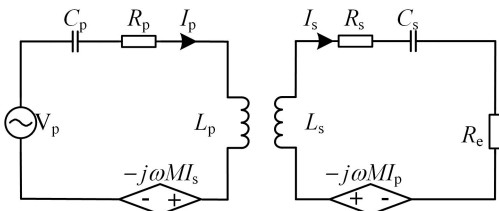

**Figure 3.** Equivalent mutual inductance model of serial–serial (SS) compensated circuit.

The subscripts "p" and "s" represent the primary side and the secondary side, respectively. The symbols $L$, $C$ and $R$ are the corresponding self-inductance, capacitance, and equivalent resistance, respectively. $M$ is coupling inductance which is equal to $k\sqrt{L_\mathrm{p}L_\mathrm{s}}$. $V_\mathrm{p}$ is the voltage generated by power inverter, and $R_\mathrm{e}$ is the equivalent resistance seen from rectifier.

By using the fundamental harmonic approximation (FHA) method and Kirchhoff's voltage low (KVL), the following matrix equation can be obtained [29].

$$
\begin{bmatrix} Z_\mathrm{p} & -j\omega M \\ -j\omega M & Z_\mathrm{s} \end{bmatrix} \begin{bmatrix} I_\mathrm{p} \\ I_\mathrm{s} \end{bmatrix} = \begin{bmatrix} V_\mathrm{p} \\ 0 \end{bmatrix} \tag{2}
$$

where, $Z_\mathrm{p} = j\omega L_\mathrm{p} + 1/(j\omega C_\mathrm{p}) + R_\mathrm{p}$ and $Z_\mathrm{s} = j\omega L_\mathrm{s} + 1/(j\omega C_\mathrm{s}) + R_\mathrm{s} + R_\mathrm{e}$. According to (1) and (2), the root-mean-square (RMS) value of primary resonant current $I_\mathrm{p}$ and secondary resonant current $I_\mathrm{s}$ are given by

$$
\begin{cases} I_\mathrm{p} = \dfrac{(R_\mathrm{s} + R_\mathrm{e}) V_\mathrm{p}}{\omega^2 M^2 + R_\mathrm{p}(R_\mathrm{s} + R_\mathrm{e})} \\[3ex] I_\mathrm{s} = \dfrac{j\omega M V_\mathrm{p}}{\omega^2 M^2 + R_\mathrm{p}(R_\mathrm{s} + R_\mathrm{e})}. \end{cases} \tag{3}
$$

According to (3), the input power $P_{in}$ from the full-bridge inverter and output power $P_{out}$ of the secondary compensation circuit are calculated as (4) and (5), respectively.

$$P_{in} = \Re\left[V_p(I_p)^*\right] = \frac{(R_s + R_e)\,V_p^2}{\omega^2 M^2 + R_p\,(R_s + R_e)} \tag{4}$$

$$P_{out} = \Re\left(R_e|I_s|^2\right) = \frac{R_e(\omega M)^2 V_p^2}{\left[\omega^2 M^2 + R_p\,(R_s + R_e)\right]^2} \tag{5}$$

where $\Re(\cdot)$, $(\cdot)^*$ and $|\cdot|$ represent the real, conjugate and absolute variables, respectively. Taking the ratio of (5) and (4), the system efficiency $\eta$ is calculated as

$$\eta = \frac{P_{out}}{P_{in}} = \frac{\omega^2 M^2 R_e}{\omega^2 M^2 (R_p + R_e) + R_s(R_p + R_e)^2}. \tag{6}$$

For a well-designed WPT system, the values of $\omega$, $M$, $R_p$ and $R_s$ are usually fixed [30], and the input power and output power are mainly determined by $V_p$ and $R_e$, respectively. It can be seen from (6) that the system efficiency is related to $V_p$, but it is determined by $R_e$. Thus, the modulation of input voltage $V_p$ has the potential to realize forward communication while the modulation of the equivalent resistance $R_e$ can realize backward communication.

### 2.3. Transient Analysis of Power and Information Channels

In the previous Section, the steady-state characteristics of the SS compensated WPT system are analyzed. However, if the SWPIT system switches from the normal power transmission mode to the simultaneous power and information transmission mode, the steady-state circuit model cannot be used to analyze the behavior of the resonant tank. By using the reciprocity theorem, the characteristics of forward and backward communication are identical. Therefore, in this Section, only the backward communication is studied, and its equivalent circuit is shown in Figure 4a.

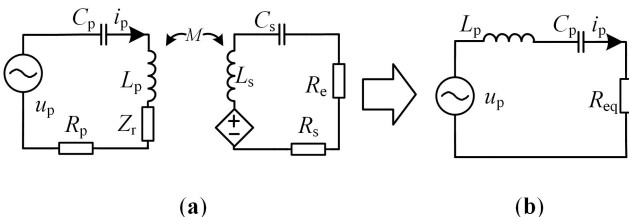

(**a**)          (**b**)

**Figure 4.** Equivalent circuit for transient analysis. (**a**) Full-order reflected impedance circuit model. (**b**) Simplified equivalent circuit model.

During the backward modulation, the resonant tank departs from its steady state. For simplicity, it is assumed that the secondary compensation circuit is in the resonant state whereas the primary compensation circuit is in the non-resonant state. Then, the full-order reflected impedance circuit in Figure 4a can be simplified as the second-order equivalent circuit model shown in Figure 4b, where the reflected impedance $Z_r$ seen from the primary side is given by

$$Z_r = \frac{\omega^2 M^2}{Z_s}. \tag{7}$$

where $Z_s = j\omega L_s + 1/(j\omega C_s) + R_p + R_e$, and the equivalent resistance seen from secondary side is $R_e$. If the WPT system works in the resonant state, the reflection impedance has pure resistive characteristic.

To analyze the simplified second-order circuit shown in Figure 4b, the equivalent impedance $R_{eq}$ is derived as

$$R_{eq} = R_p + \frac{(\omega M)^2}{R_s + R_e}. \tag{8}$$

For simplicity, the initial communication process is regarded as zero-voltage response [31]. Then, the differential equation of the simplified circuit shown in Figure 4b is expressed as

$$L_p C_p \frac{d^2 i_p(t)}{dt^2} + R_{eq} C_p \frac{d i_p(t)}{dt} + i_p(t) = 0. \tag{9}$$

Solving (9), the instantaneous current $i_p$ in primary side is derived as

$$i_p = \frac{-v_c(0)}{\omega L_p} e^{-t/\tau} \sin \omega t + i_p(0) \frac{\omega}{\omega_0} e^{-t/\tau} \cos(\omega t + \theta). \tag{10}$$

where $v_c(0)$ and $i_p(0)$ are initial voltage on $C_p$ and initial current in $L_p$, respectively. The relationship between attenuation angular frequency and operation angular frequency is $\omega_0 = \omega \sqrt{1 - \xi^2}$, in which $\omega = 1/\sqrt{L_p C_p}$. In order to decrease the transient response time, the equivalent circuit is designed in an under-damped state and $\xi < 1$. Assuming that the switching point of the phase-shifting angle always occurs at zero point, then the value of $i_p(0)$ is zero. At the same time, $v_c(0)$ reaches its maximum value, then (10) can be simplified as

$$i_p = \frac{v_p(t) + v_c(0)}{\omega L_p} e^{-t/\tau} \sin \omega t. \tag{11}$$

where, $\tau = 2L_p/R_{eq}$. According to (8) and (11), the transient response time $\tau$ of primary current is calculated as

$$\tau = \frac{2L_p}{R_p + \frac{(\omega M)^2}{R_s + R_e}}. \tag{12}$$

In a well-designed WPT system, the equivalent resistances of coils satisfy $R_p \approx R_s \ll R_e$. As evident from (12), in order to achieve high data rate in the communication process, the value of $L_p$ and $R_e$ should decrease, and/or the value of $\omega$ and $M$ should increase.

## 3. Communication System Design

### 3.1. Forward Communication Based on Full-Bridge Inverter

On the primary side, a full-bridge inverter that works in phase-shifting mode is employed to regulate input power for resonant tank, as shown in Figure 5a. The output voltage and output current are defined as $V_{inv}$ and $i_p$, respectively. In terms of the modulator, as shown in Figure 5b, the main counter and the shift counter are used to generate digital count values, and they share a same clock source (CLK). Comparing the value of the two counters with the half count value will generate a square wave signal with 50% duty cycle. Unlike the main counter, the initial value of the shift counter is determined by the transmitted data shown in (15). Finally, the obtained four signals (PWM$_1 \sim$ PWM$_4$) are used to drive the four MOSFETs ($Q_1 \sim Q_4$), respectively. The output voltage of full-bridge inverter with Fourier series form is expressed as

$$V_{inv} = V_{dc} \frac{4}{\pi} \sum_{n=1,3,\ldots}^{\infty} \frac{1}{n} \cos\left(n\omega t + \frac{n\varphi_p}{2}\right) \sin\left(\frac{n\varphi_p}{2}\right). \tag{13}$$

where, $n$ represents the number of harmonic components, and $V_{dc}$ is the voltage on the DC bus. $\omega$ is the operation angular frequency of WPT system, and $\varphi_p$ is the phase-shifting angle of the full-bridge inverter. For the SS compensated WPT system, the resonant tank composed of compensation circuits

and coils is regarded as a band-pass filter, and only the fundamental component of $V_{\text{inv}}$ is considered in the power loop. As a result, Equation (13) can be simplified as Equation (14).

$$V_{\text{inv},1} = V_{\text{dc}} \frac{4}{\pi} \cos\left(\omega t + \frac{\varphi_{\text{p}}}{2}\right) \sin\left(\frac{\varphi_{\text{p}}}{2}\right) \tag{14}$$

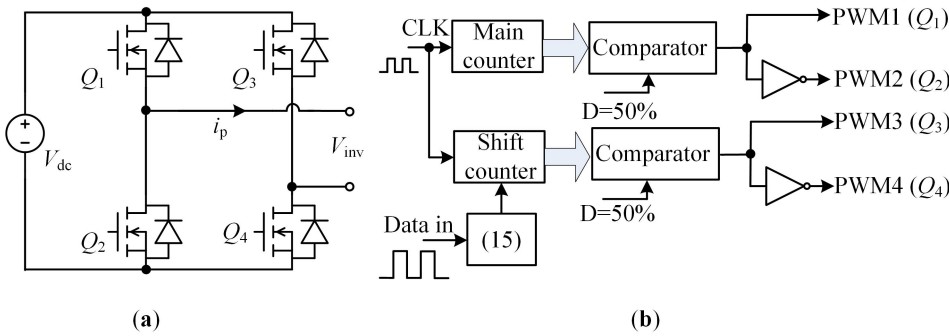

(**a**)  (**b**)

**Figure 5.** (**a**) Circuit of full-bridge inverter. (**b**) Block diagram of modulation system based on differential phase shift keying (DPSK).

From (14), the output voltage of inverter is regulated by $V_{\text{dc}}$ and $\varphi_{\text{p}}$. In practice, the value of $V_{\text{dc}}$ is constant, and $\varphi_{\text{p}}$ will vary with different power demand and coupling inductance. Here, a differential phase-shifting angle $\Delta\varphi_{\text{p}}$ is added into $\varphi_{\text{p}}$, so that the full-bridge inverter is capable of information transmission. Taking binary symbol communication as an example, the changes of phase-shifting angle during the modulation of bit "1" and bit "0" is given by (15).

$$\begin{cases} \varphi_{\text{p}} - \Delta\varphi_{\text{p}} \to \varphi_{\text{p}} + \Delta\varphi_{\text{p}} & \text{"1"} \\ \varphi_{\text{p}} & \text{"0"} \end{cases} \tag{15}$$

When bit "0" is sent, the phase-shifting angle maintains its initial value for two switching cycles. When bit "1" is sent, the phase-shifting angle is set to $\varphi_{\text{p}} - \Delta\varphi_{\text{p}}$ in the first switching cycle, and then becomes $\varphi_{\text{p}} + \Delta\varphi_{\text{p}}$ in the next switching cycle. Since the change of phase-shifting angle is differential and complementary, the modulation method is defined as differential phase shifting keying (DPSK).

To ensure the stability of power transmission, the value of $\Delta\varphi$ should be designed as small as possible. However, the detection of information transmission depends on the fluctuation of secondary current, which requires higher value of $\Delta\varphi$. Therefore, the value of $\Delta\varphi$ is determined by the reliability of communication as well as the performance of power transmission. To further illustrate the working principle of the modulator designed above, bit "1" and bit "0" are sent, and the corresponding waveforms of the full-bridge inverter and modulator are depicted in Figure 6.

When considering the fundamental component of the power loop, the output voltage of the full-bridge inverter satisfies $V_{\text{p}} = V_{\text{inv}}$. Then, the average input power of power loop is given by

$$P_{\text{in}} \approx \frac{V_{\text{p}}^2}{R_{\text{eq}}}. \tag{16}$$

Substituting (4), (8), (14) and (15) into (16), the relation of average input power during the modulation of bit "1" and bit "0" is derived as

$$P_{\text{in}}(\varphi_{\text{p}} - \Delta\varphi_{\text{p}}) + P_{\text{in}}(\varphi_{\text{p}} + \Delta\varphi_{\text{p}}) \approx 2P_{\text{in}}(\varphi_{\text{p}}). \tag{17}$$

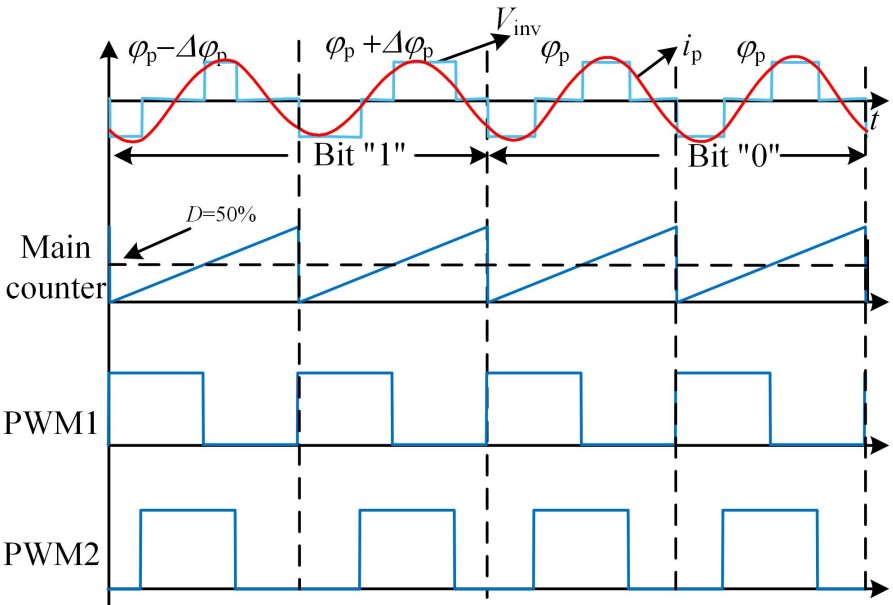

**Figure 6.** Key waveforms of full-bridge inverter and modulator during the modulation of bit "1" and bit "0".

Obviously, the modulation of bit "0" is equivalent to the non-communication mode. It can be seen from (17) that the average input power during modulating bit "1" is almost equal to that in the modulation of bit "0" for every two switching cycles. Therefore, the forward communication with DPSK modulation has little impact on the power transmission.

### 3.2. Backward Communication Based on S-BAR

Compared with passive rectifier and full-bridge active rectifier, S-BAR has fewer switching devices and more regulating degrees [32]. For the S-BAR, synchronization signal plays an important role in power regulation. Usually, the phase of resonant current in compensation circuit serves as synchronization signal, which is measured by a current sensor on the secondary side, as shown in Figure 7. To eliminate the noise caused by hard switching and dead time, a low-pass filter (LPF) is used. However, both the current sensor and the LPF will cause phase delay. Therefore, a phase leading correction is required to eliminate the phase delay. In this paper, the BPF and zero-crossing detection are realized through analog circuit, and the phase leading correction is realized through FPGA. There are two reasons for choosing FPGA: (1) The leading phase can be easily adjusted according to the operation frequency. (2) It can be integrated into the modulator on the secondary side, therefore, the hardware cost of SWPIT system can be reduced.

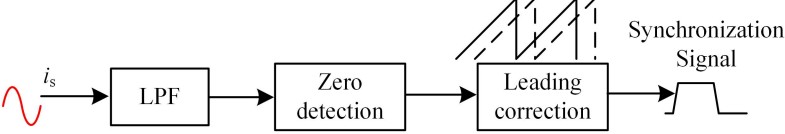

**Figure 7.** Synchronization circuit for semi-bridgeless active rectifier (S-BAR).

Similar to the modulation of forward communication, the data carrier and power carrier are also shared in the backward communication. Then, the phase-shifting angle $\varphi_s$ is used to regulate output power and equivalent resistance, which is determined by the modulation of bit "1" and bit "0". The changes of phase-shifting angle during the modulation of bit "1" and bit "0" is given by

$$\begin{cases} \varphi_s - \Delta\varphi_s \rightarrow \varphi_s + \Delta\varphi_s & \text{"1"} \\ \varphi_s & \text{"0".} \end{cases} \tag{18}$$

Figure 8b shows the block diagram of modulation based on DPSK, in which two digital counters with 12 bit data-width are used to record the value of source clock (CLK). The counter 1 and counter 2 recount at the rising and falling edges of synchronization signal, respectively. By this way, the positive and negative period of secondary current is measured separately. Meanwhile, the value of $\varphi_s$ and $\Delta\varphi_s$ are determined by (18). Once the value of counter is between $\varphi_s$ and $\pi - \varphi_s$, the corresponding MOSFET turns OFF. Otherwise, it turns ON. The detailed control strategy for MOSFETs ($Q_5 \sim Q_6$) and diodes ($D_1 \sim D_2$) is listed in Table 1.

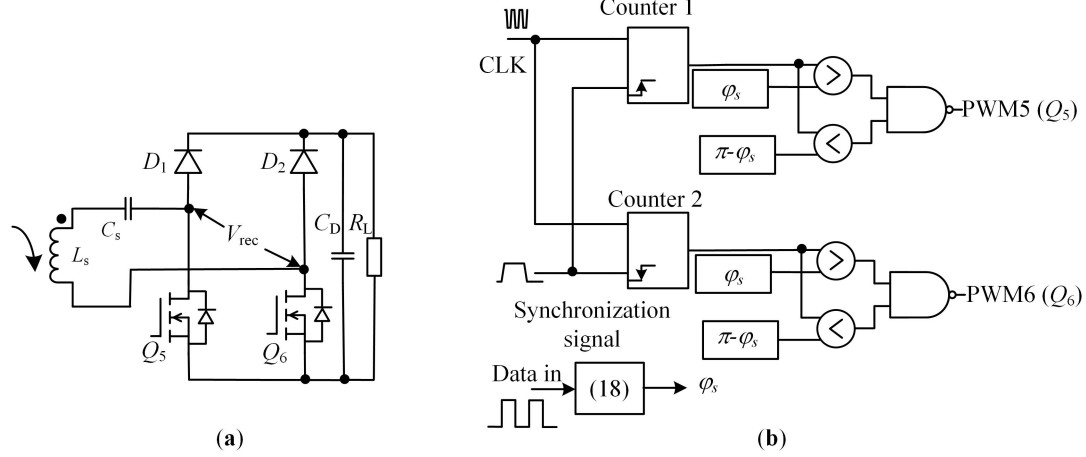

(**a**)　　　　　　　　　　　　　　　　　　　　　　　　　　　　(**b**)

**Figure 8.** (**a**) Circuit of semi-bridgeless active rectifier (S-BAR). (**b**) Block diagram of DPSK based modulation.

**Table 1.** Control strategy for power devices during one switching cycle.

| Periods | MOSFETs | Diodes | Power Transmission |
|---------|---------|--------|--------------------|
| $t_0$-$t_1$ | $Q_5$ is ON, $Q_6$ is ON | $D_1$ is OFF, $D_2$ is OFF | Within resonant tank |
| $t_1$-$t_2$ | $Q_5$ is OFF, $Q_6$ isON | $D_1$ is ON, $D_2$ is OFF | To load |
| $t_2$-$t_3$ | $Q_5$ is ON, $Q_6$ is ON | $D_1$ is OFF, $D_2$ is OFF | Within resonant tank |
| $t_3$-$t_4$ | $Q_5$ is ON, $Q_6$ is ON | $D_1$ is OFF, $D_2$ is OFF | Within resonant tank |
| $t_4$-$t_5$ | $Q_5$ is ON, $Q_6$ is OFF | $D_1$ is OFF, $D_2$ is OFF | To load |
| $t_5$-$t_6$ | $Q_5$ is ON, $Q_6$ is ON | $D_1$ is OFF, $D_2$ is OFF | Within resonant tank |

Obviously, there are two periods when the load obtains energy, otherwise the energy is blocked by MOSFETs. Then, the equivalent resistance seen from S-BAR changes dynamically according to $\varphi_s$. To further understand the backward communication, the waveforms of the S-BAR and modulator are depicted in Figure 9 during the modulation of bit "1" and bit "0".

According to the phase relation of secondary voltage $V_{rec}$ and current $i_s$, the S-BAR has three working modes, i.e., capacitive, inductive, and resistive modes [33]. To minimize the power losses caused by hard switching, the primary voltage should slightly lead the primary current to realize zero voltage switching (ZVS) [34]. As a result, the S-BAR should be designed in capacitive or resistive mode, and the equivalent impedance $Z_e$ seen from S-BAR is written as

$$Z_e = R_e + jX_e. \tag{19}$$

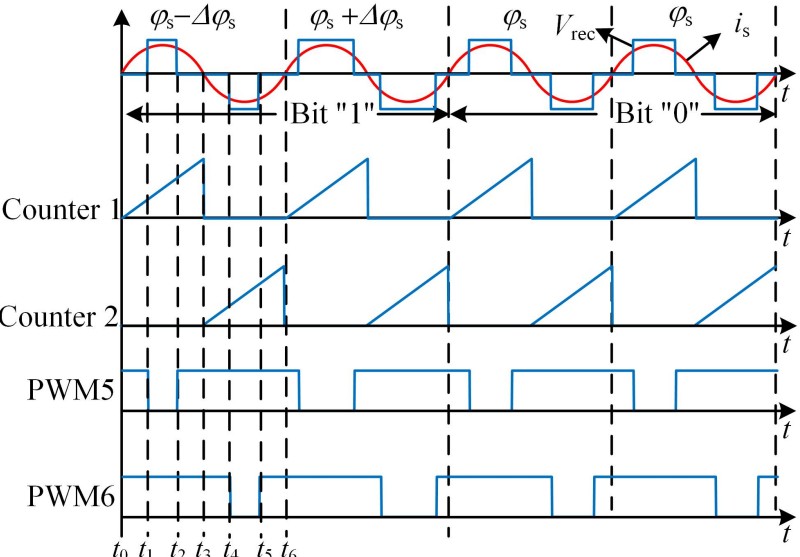

**Figure 9.** Key waveforms of the S-BAR and modulator during the modulation of bit "1" and bit "0".

In the capacitive mode, the real and imaginary parts of $Z_e$ are given by

$$\begin{cases} R_e = \Re(Z_{eq}) = \dfrac{8}{\pi^2} R_L \sin^4 \left( \dfrac{\varphi_s}{2} \right) \\ \\ X_e = \Im(Z_{eq}) = \dfrac{8}{\pi^2} R_L \sin^3 \left( \dfrac{\varphi_s}{2} \right) \cos \left( \dfrac{\varphi_s}{2} \right). \end{cases} \tag{20}$$

In the resistive mode, the imaginary part is zero and the real part is given by

$$R_e = \Re(Z_{eq}) = \frac{8}{\pi^2} R_L \sin^2 \left( \frac{\varphi_s}{2} \right). \tag{21}$$

Whether the secondary side is capacitive or resistive mode, the current in secondary compensated circuit has constant characteristic. Substituting (20) or (21) into (5), the relation of output power during the modulation of bit "1" and bit "0" is derived as

$$P_{out}(\varphi_s - \Delta\varphi_s) + P_{out}(\varphi_s + \Delta\varphi_s) \approx 2P_{out}(\varphi_s). \tag{22}$$

As evident from (22), the output power in the modulation of bit "1" is almost equal to that in the modulation of bit "0" during every two switching cycles. Therefore, the backward communication with DPSK modulation has little impact on power transmission.

*3.3. Demodulator Design*

The data demodulation is dependent on the envelope characteristics of the primary and secondary side currents, therefore, the demodulators on both sides can be designed with a similar structure. According to the functionality, the demodulator circuit shown in Figure 10 can be divided into four parts: current to voltage ($I/V$) converter, precision full-wave rectifier, passive third-order low-pass filter and comparator. For the $I/V$ converter, its input voltage $V_{in}$ is determined by the ratio of high-frequency current sensor and the value of $R_x$, and the transconduction gain can be expressed as $G_{iv} = KR_x i_s$. In practice, the variation of voltage of current sensor is too small, and it is easily

overwhelmed by the threshold voltage of diode. Then, the precision full-wave rectifier will compensate this voltage, and the output voltage on rectifier is given by

$$V_{\text{rec,dc}} = \frac{4V_{\max}}{\pi} \left[ \frac{1}{2} + \frac{1}{1 \times 3} \cos{(2\omega t)} - \frac{1}{3 \times 5} \cos{(4\omega t)} + \frac{1}{5 \times 7} \cos{(6\omega t)} - ... \right]. \tag{23}$$

where, $V_{\max}$ is the maximum value of $V_{\text{rec,dc}}$. It can be clearly seen from (23) that the voltage $V_{\text{rec,dc}}$ consists of even-order components and DC component. In order to eliminate these components, a third-order low-pass filter is used.

Notably, the voltage from demodulator should be designed high enough for the comparator, and the SNR should be increased to improve communication performance. Ignoring the non-linear characteristics of current sensor and diodes, the transfer function of demodulator is given by

$$G(s) = G_{\text{iv}} \frac{R_6}{C_1 C_2 L_1 R_5 R_6 s^3 + (C_1 L_1 R_5 + C_2 L_1 R_6) s^2 + (C_1 R_5 R_6 + C_2 R_5 R_6 + L_1) s + (R_5 + R_6)}. \tag{24}$$

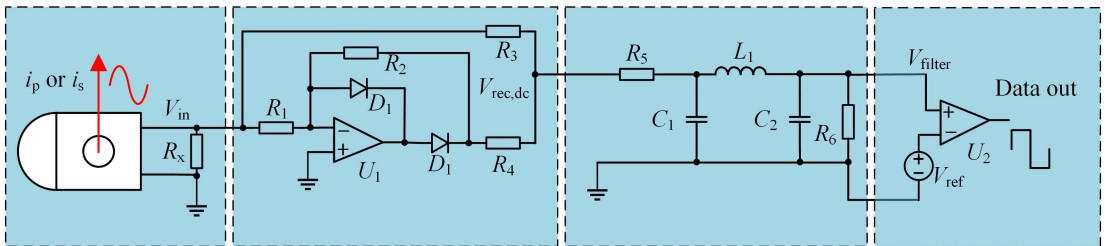

**Figure 10.** Circuit of demodulator.

By selecting appropriate component value, the Bode plot of demodulator is calculated by MATLAB, as shown in Figure 11. It has low magnitude attenuation in the low frequency domain, so it can extract low frequency current fluctuations that carry information. Meanwhile, the fundamental and high-order components are blocked at the cut-off frequency ($f_c$=200 kHz).

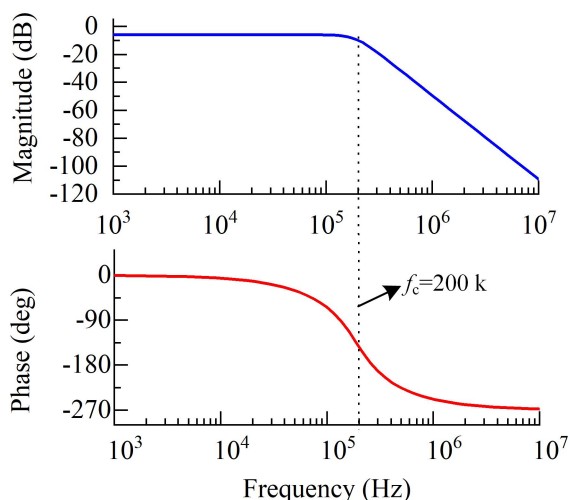

**Figure 11.** Bode plot of the designed demodulator.

### 3.4. Half-Duplex Communication Protocol

In practice, the variations of input voltage, coupling inductance and load resistance are inevitable, leading to changes in primary and secondary current. Moreover, the data carrier and power carrier of forward and backward communication are shared, and there is an interaction between power and

information. Therefore, it is necessary to design a half-duplex protocol to enhance communication performance. In the protocol, the length of data frame should be designed as small as possible to achieve high data rate [35]. However, if it is too small, error bits may occur during communication. According to the priority of information exchange, there are two types of protocols for practical application, i.e., command protocol and data protocol. The definition of these two protocols are listed in Tables 2 and 3, respectively. Both of them start with bit "1" and end with bit "1". To ensure the validity of the transmitted bits, a parity bit is inserted before the stop bit to detect the erroneous bit. Then, four bits representing address matching are added to improve the interoperability for different WPT systems. The total length of command protocol is only eleven, which will facilitate emergency information transmission, such as handshake between devices, device failure, protection information, etc. For the data protocol listed in Table 3, four bits are required to define the type of data, such as voltage, current, power, etc. In order to increase the data rate, the data representing the values of different type information has only ten bits, which is large enough for typical high-resolution voltage and current sensor.

**Table 2.** Definition of command protocol.

| Start Bit | Address | Command | Parity Bit | Stop Bit |
|-----------|---------|---------|------------|----------|
| 1 bit | 4 bit | 4 bit | 1 bit | 1 bit |

**Table 3.** Definition of data protocol.

| Start Bit | Address | Data Type | Data | Parity Bit | Stop Bit |
|-----------|---------|-----------|------|------------|----------|
| 1 bit | 4 bit | 4 bit | 10 bit | 1 bit | 1 bit |

Figure 12a shows the detailed flowchart of half-duplex communication, where "Tx" and "Rx" represent the sending and receiving data modes, respectively. These two modes are determined by the initial current in primary or secondary side. If current fluctuation is detected, "Rx" mode is selected. The transmitted data is unpacked with protocol above then the beginning state occurs until the received bits are completely processed. If there is a command or data protocol to be sent and the initial current is stable, the "Tx" mode is selected. After all bits are packed with protocol and completely sent, the beginning state occurs and waits for the next communication process. The working sequence is shown in Figure 12b, where the "x" and "Rx" modes work in turn on both side, and these two modes can be regarded a cycling communication process.

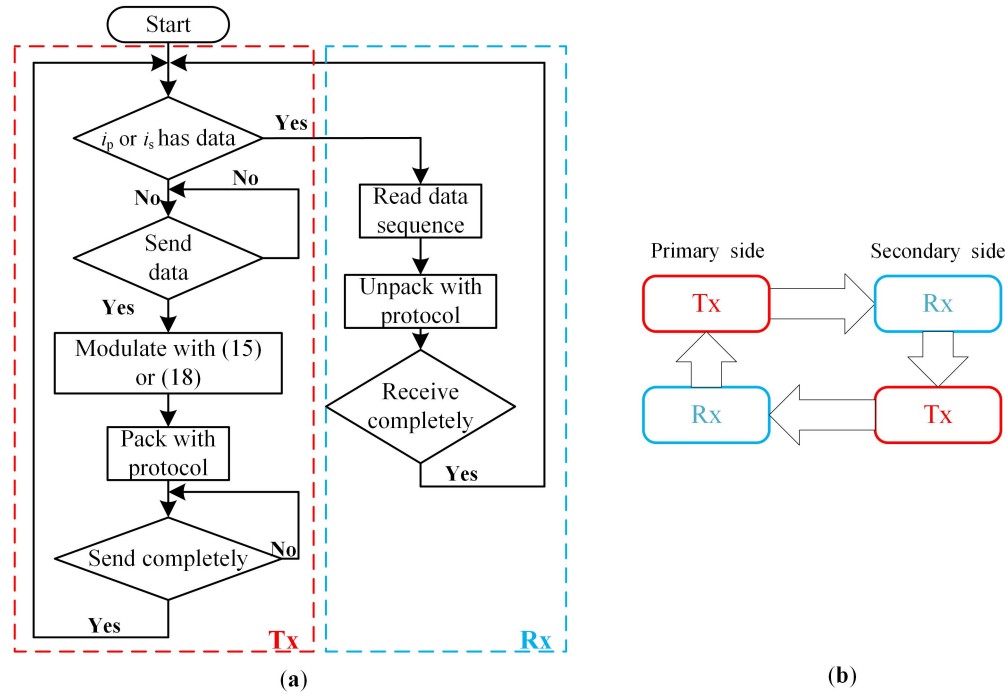

**Figure 12.** Diagram of the half-duplex communication. (**a**) Flowchart of "Tx" and "Rx" on primary and secondary side. (**b**) Operation sequence of SWPIT system.

## 4. Simulation and Experiment

### 4.1. Simulation

To verify the correctness of the proposed SWPIT system, the forward and backward communication are studied through the PLECS software. The overall schematic diagram is depicted in Figure 13. The simulation step of the solver is set to 20 ns to ensure the accuracy of simulation results. The modulators and demodulators are implemented by C-scribe block and analog circuit, respectively. The parameters related to the simulation are listed in Table 4.

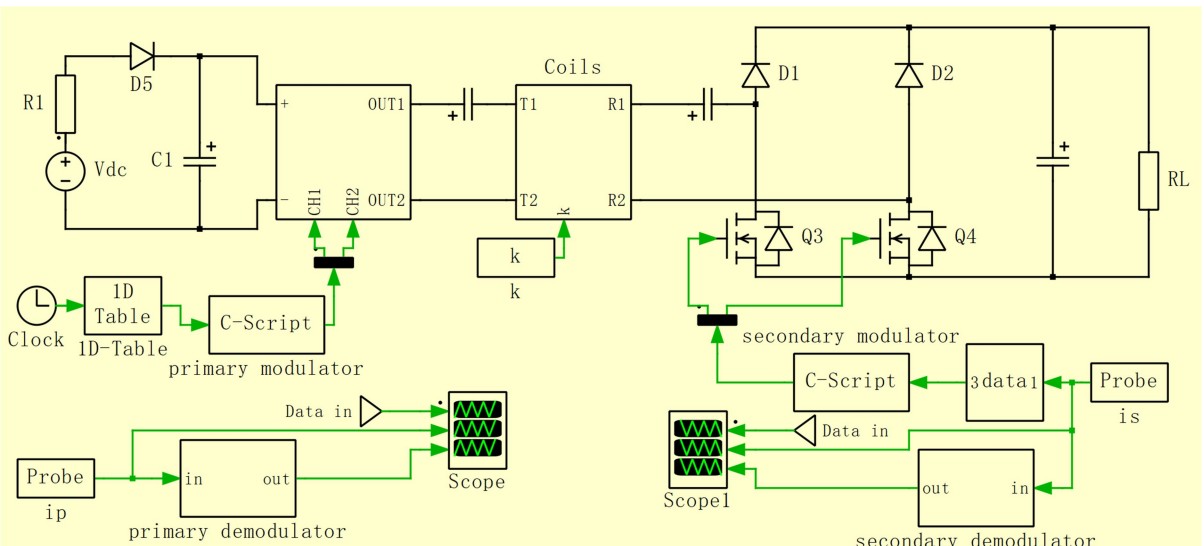

**Figure 13.** Overall diagram of simulation circuit.

For the forward communication, the transmitted bits ("1010...") with the data rate of 25 kbps are loaded on the primary side and mixed with the power modulator. After about three switching periods,

the transmitted bits are recovered by the demodulator. The related waveforms of secondary current $i_s$, transmitted bits and received data are shown in Figure 14a.

Compared with the forward communication, the transmitted bits of backward communication are the same but the data rate is different, as shown in Figure 14b. Due to the constant current characteristic in the SS compensated circuit, the fluctuation of the primary current $i_p$ is smaller than the secondary current $i_s$.

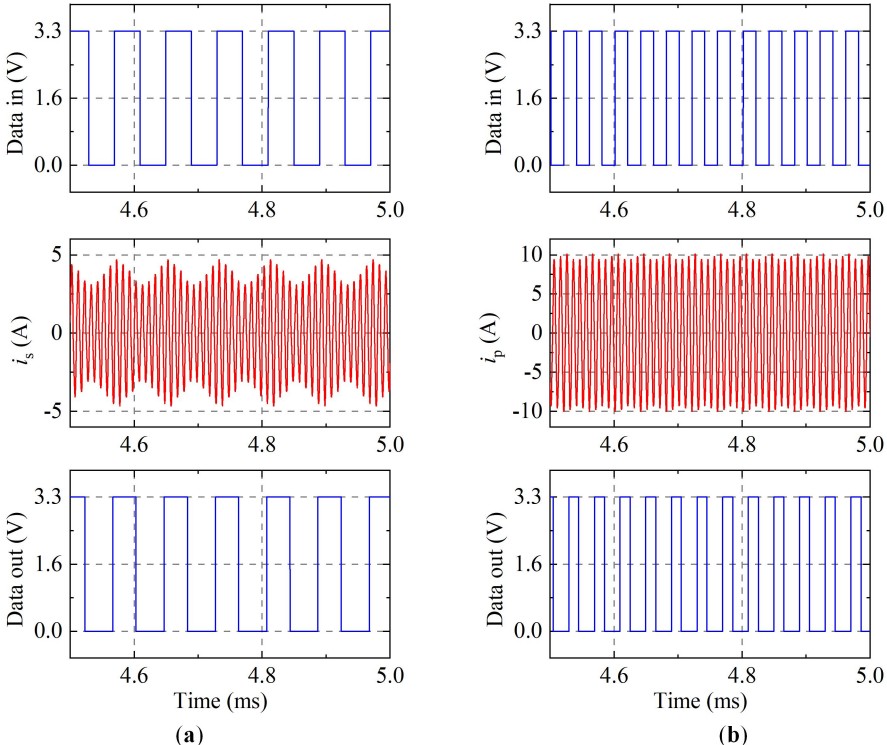

**Figure 14.** Simulated waveforms of transmitted bits (Data in), resonant currents ($i_p$ and $i_s$) and received bits (Data out). (**a**) Forward communication with 25 kbps data rate. (**b**) Backward communication with 50 kpbs data rate.

**Table 4.** Parameters of simulation and experiment.

| Parameters | Description | Simulation | Experiment |
|:---:|:---|:---:|:---:|
| $f$ | Operating frequency | 100 kHz | |
| $V_{dc}$ | Voltage of DC power source | 60 V | |
| $R_L$ | Load resistance | 10–25 Ω | |
| $k$ | Coupling coefficient | 0.1–0.3 | |
| $L_p$ | Primary coil inductance | 50 μH | 49.2 μH |
| $C_p$ | Compensated capacitor of primary side | 50 nF | 51.4 nF |
| $L_s$ | Secondary coil inductance | 50 μH | 50.4 μH |
| $C_s$ | Compensated capacitor of secondary side | 50 nF | 49.7 nF |
| $R_p$ | Equivalent resistance of primary coil | 0.1 Ω | 0.18 Ω |
| $R_s$ | Equivalent resistance of secondary coil | 0.1 Ω | 0.18 Ω |

### 4.2. Experimental Prototype

To verify the effectiveness of the proposed DPSK communication method, a 350 W experimental prototype is established, as shown in Figure 15a. The related experimental parameters are listed in

Table 4. The coils of primary and secondary side are identical, which are wounded with $\Phi 0.07$ mm $\times$ 350-stranded Litz wire. To concentrate magnetic field, six ferrite strips are fixed by a plastic structure. The full-bridge inverter and the low-side of S-BAR are constructed by SiC MOSFETs (C2M0080120D) while the high-side of S-BAR is constructed by fast-recovery diodes (DSEI120-06A). The DPSK based modulators of primary and secondary sides are implemented by two FPGAs (EP4CE6E22CBN). An electric load is employed to dynamically adjust the value of load resistance. Figure 15b illustrates the circuit of demodulator on both sides, in which, according to the required power demand, an adjustable resistor is used to regulate the reference value for the comparator. In the experiment, all waveforms are captured by an oscilloscope (RIGOL MSO1104Z).

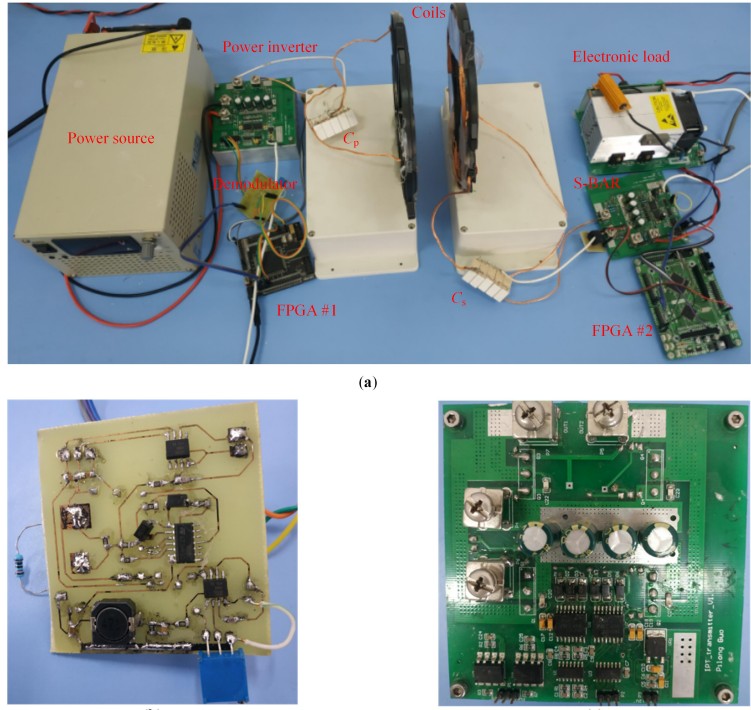

**Figure 15.** Photograph of the experimental prototype. (**a**) Overall configuration of experimental prototype. (**b**) Demodulator circuit. (**c**) S-BAR circuit.

*4.3. Experimental Results*

Figure 16a shows the measured output voltage and current of the full-bridge inverter when the transmitted bits (Data in) are "010". The initial phase-shifting angle $\varphi_\mathrm{p}$ and differential phase-shifting angle $\Delta\varphi_\mathrm{p}$ are set to $90°$ and $30°$, respectively. The variations of $\varphi_\mathrm{p}$ and $\Delta\varphi_\mathrm{p}$ are consistent with the variations in Figure 6. In the forward modulation process, $i_\mathrm{p}$ is almost in the phase of $V_\mathrm{inv}$, and zero phase angle of inverter is achieved. Figure 16b shows the corresponding waveforms of $i_\mathrm{s}$, $V_\mathrm{filter}$ and received bits (Data out). The transmitted bits are successfully recovered from the demodulator on the secondary side.

Figure 17a shows the measured input voltage and current of the S-BAR when the transmitted bits are "010". It can be seen that $i_\mathrm{s}$ slightly leads $V_\mathrm{rec}$ during the backward modulation but its peak value is almost stable. Similar to the forward communication, the transmitted bits are recovered from the demodulator on the primary side.

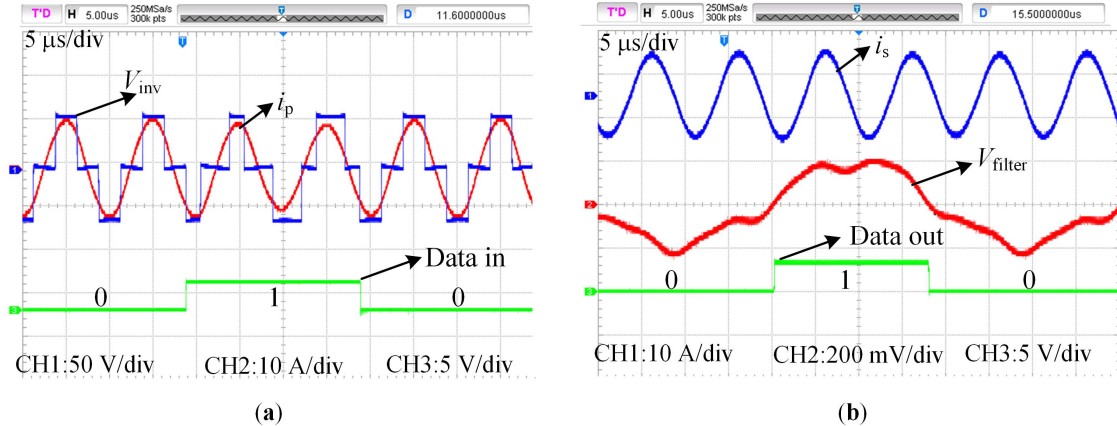

**Figure 16.** Experimental results of forward communication. (**a**) Measured waveforms of $V_{inv}$ and $i_p$ when the transmitted bits are "010" (Data in) under $\varphi_p = 90°$. (**b**) Measured waveforms of $i_s$, $V_{filter}$ and received data (Data out).

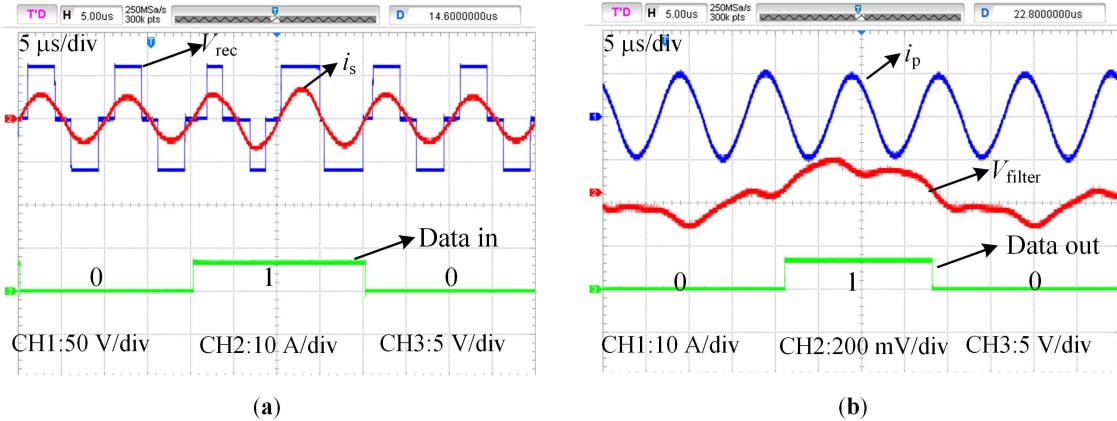

**Figure 17.** Experimental results of backward communication. (**a**) Measured waveforms of $V_{rec}$ and $i_s$ when the transmitted bits are "010" (Data in) under $\varphi_s = 80°$. (**b**) Measured waveforms of $i_p$, $V_{filter}$ and received data (Data out).

To further verify the performance of the designed communication system, a bit frame "10001000110" produced by a random bit sequence generator is sent from primary side to the secondary side. After about 116 µs, the transmitted bits are successfully recovered from secondary side, as shown in Figure 18a. During the forward communication, the maximum fluctuation of $V_{out}$ is about 300 mV at its rated output voltage 24 V, as shown in Figure 18b, which is acceptable in practice. Thus, the information transmission has little impact on power transmission.

Figure 19a compares the DC to DC efficiencies with and without communication under different coil distance. As the distance increases, $\eta$ gradually decreases but the trends of three different conditions are similar. Because the modulation of backward communication changes the equivalent resistance of load, the system efficiency of backward communication is slightly lower than that of forward communication. Generally, the impact of the communication process on the system efficiency can be negligible. Figure 19b compares the measured bit error rate (BER) under different data rates (50 kbps, 25 kbps and 12.5 kbps) during the forward communication. As the distance increases, the BER under three rates rises significantly. Under the same distance, the BER at lower data rates is relatively small.

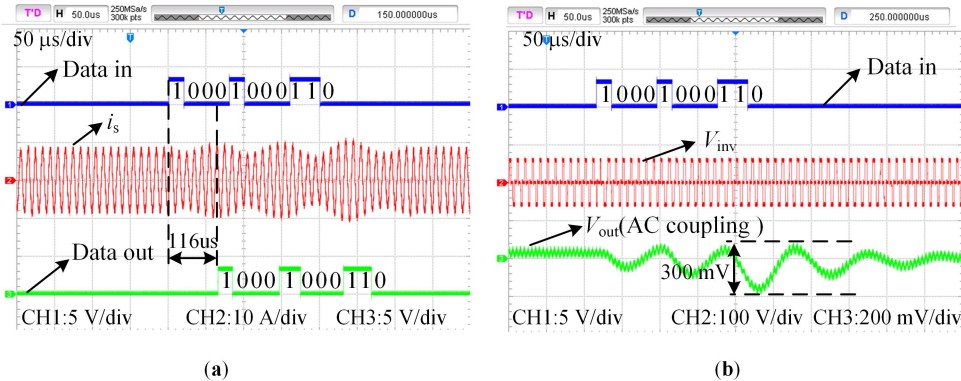

**Figure 18.** Measured waveforms of forward communication under the bit frame "10001000110". (**a**) Transmitted data (Data in), secondary current $i_s$ and received data (Data out). (**b**) The ripple of output voltage $V_{out}$ under $k = 0.15$ and $R_L = 25\Omega$.

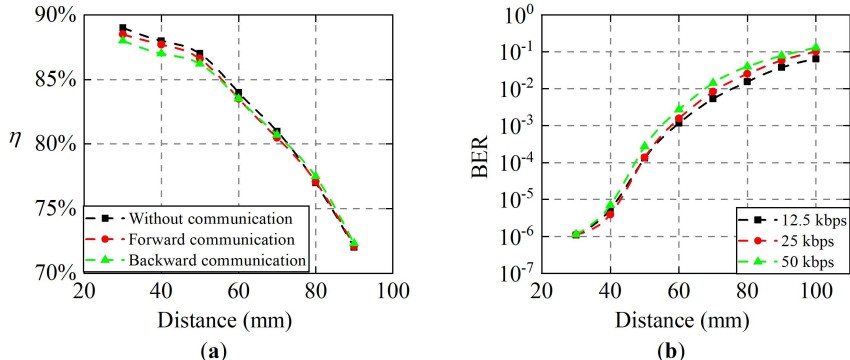

**Figure 19.** Power and information transmission under different coil distance. (**a**) System efficiency with and without communication. (**b**) Measured bit error rate (BER) under different data rates.

When $\varphi_{p,s} = 90°$, $\Delta\varphi_{p,s} = 30°$, and $R_L = 20\ \Omega$, the fluctuation of secondary current $\Delta i_s$ and primary current $\Delta i_p$ are shown in Figure 20a,b, respectively. Due to the constant current characteristic, $\Delta i_p$ in the backward communication is smaller than $\Delta i_s$ in the forward communication. Because of the nonlinear characteristics of the rectifier and scattering resistance of the coils, there is a small difference between experiment and simulation results, especially when the coupling coefficient is lower than 0.2. The experiment results almost follow the simulation results.

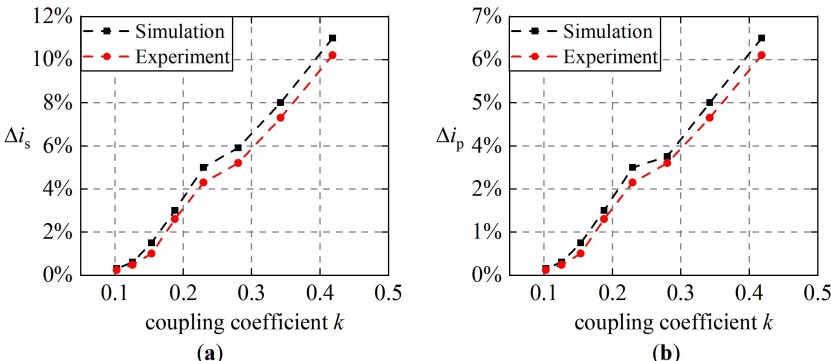

**Figure 20.** The fluctuations of secondary and primary currents with simulation and experiment under different coupling coefficients. (**a**) Forward communication. (**b**) Backward communication.

## 5. Conclusions

To maintain the stability of power transmission during information transmission, a bidirectional communication based on the DPSK modulation method was proposed in this paper. The analysis of steady-state and transient circuit models indicated the potential for simultaneous power and information transmission. Then, considering the interaction of information transmission, the primary and secondary communication modulators sharing with power modulators were designed. The demodulator and half-duplex communication protocol were implemented by analog circuit and software, respectively. Both simulation and experimental results showed that the effectiveness of the proposed DPSK communication method, and the maximum data rate reached up to 50 kbps. During the communication process, the power and information transmission was quasi-independent. In the future, the designed DPSK modulation communication module will be integrated into controller to track maximum power efficiency and regulate the power demand of load.

**Author Contributions:** R.Y., P.G., and C.C. conceptualized the main idea of this research project; P.G. and L.Y. designed and conducted the experiments; R.Y. and C.C. checked the results; R.Y. and P.G. wrote the whole paper. All authors have read and agreed to the published version of the manuscript.

**Funding:** This research received no external funding.

**Conflicts of Interest:** The authors declare no conflict of interest.

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
