# Peer review of "Quasi-Independent Bidirectional Communication Methods for Simultaneous Wireless Power and Information Transmission"

_applsci, doi:10.3390/app10207130_

Round 1

Reviewer 1 Report

No remarks. It is good and interesting paper connected with area of simultaneous wireless power and information transmission with proposition of novel bidirectional communication method based on differential phase shift keying. The structure of this paper is organized properly and level of English language is suitable. New solution are described and presented  properly. Verrification are suitable made on base of simulations and experiment.  Conclusions are coonected with performed investigations of Authors.   

Author Response

No remarks. It is good and interesting paper connected with area of simultaneous wireless power and information transmission with proposition of novel bidirectional communication method based on differential phase shift keying. The structure of this paper is organized properly and level of English language is suitable. New solution is described and presented properly. Verifications are suitable made on base of simulations and experiment.  Conclusions are connected with performed investigations of Authors.   

Response: We appreciate your hard work. We hope and believe more researchers working in related fields will be interested in this topic. And we are very glad to obtain comments from expert in related fields.

Reviewer 2 Report

The Authors ought to be congratulated for this work that spans from the design and discussion of the operating principles to the demonstration by means of a prototype of a wireless charging system, integrating data communication in the power channel.

There are some typing mistakes and English in general should be double checked. I have put some examples in the comments.

1) Lines 60-61: "Power transmission is affected by power transmission"; please, revise and amend.

2) Line 61: please, clarify the reason for instability.

3) Line 69. The verb "share" misses its subject, twice.

4) Line 68. Please, clarify "data rate is increased". The limit is always in principle the switching frequency of the converter, as for method 1(c). You mean that the throughput is higher? Please, clarify.
Also take into account your modulation that outputs a bit only every 2 cycles of the inverter waveform, whose frequency is determined by the resonance of the coupling circuit. The prototype then shows testes with 25 kHz rather than the maximum 100 kHz.
Could you anticipate the main points, so that the reader is prepared?

5) Line 79, Figure 2. The sentence speaks of the schematic diagram of the proposed converter, but Figure 2 is only a concept diagram.
Please expand it with details and a scheme.

6) Eq. (1). It means that Cp and Cs must be selected to have the same resonance; considering that there are several stray capacitance terms (the coils themselves and the semiconductors), please, clarify how tuning is done.
Second point: each circuit at primary or secondary, sees the other one though the magnetic coupling, so the resulting resonant pulsation omega should be the result of the solution of the overall circuit, including primary, secondary and mutual coupling.
See also comment for Figure 4.

7) Figure 4. It is not clear how the circuit on the left can be simplified into the circuit on the right. Inductance Ls and capacitance Cs disappear, whereas they should be combined together.
See also comment for Eq (1).

8) Line 111, below eq (12). Typing mistake: Rp = Rp; probably it is Rs.

9) Below Eq (13). "composing"should be "composed"

10) Line 164, above Fig. 10. Typing mistake: "caparator"

11) Above eq (23). You write "using the full-wave rectifier circuit that compensates this voltage", but looking at Fig. 10 what you have done is using a zero-threshold diode rectifier, built around an ope. amp. Please, clarify in the text.

12) Table 2 and Table 3. Adress => Address

Reviewer 3 Report

In the paper, the Authors deal with highly important problem of simultaneous power and information transfer by wireless link. In the proposed construction, They implemented and developed the latest achievements that have recently appeared in the subject literature. Although the idea of using the differential phase shift keying to bidirectional communication in wireless power transfer systems has been previously launched, the Authors presented its principles in very clear and attractive way and it should be interested for potential readers. The paper is well prepared. It includes comprehensive introduction with the expanded background. The analytical and design considerations were implemented in practice, both in simulation studies by using PLECS software as well as in experimental prototype that justified the assumption of power transmission stability while the forward and backward communication is performed. Text is written carefully, only minor mistakes can be found.

I wonder if it is possible to compare simulation results with measurements, e.g. in regards with the fluctuation of the currents or system efficiency, especially that there is the difference between the values of arameters/passive components that are assumed in the simulation and that are implemented in the prototype. It could be interesting to see whether the real electronic system follows the simulation model. Such a comparison would confirm the professionalism of the authors / designers - model first and then time-consuming sample praparation with no surprises. 

Minor remarks:

- Please check spelling the Authors’ names (especially the first one)

- “The power transmission is easily affected by the power transmission“ – this sequence should be corrected

- Since the considerations in sections 2.2. – 2.3 are derived on the basis of the subject literature the Authors should insert more adequate references.

- the adjective “same” is used not properly in a few places of the text

- “a 350 W” – should not be divided into separate lines of the text

Round 2

Reviewer 2 Report

The Authors must be thanked for their extensive and clarifying replies. I agree with the applied changes and I have no other comments on the manuscript, that since the first version was in a good shape and interesting.

I suggest only two points regarding the replies to two of my comments and then the extension of the subsequent implementation in the paper.

Comment 4). Your explanation of the increase of transient response and necessary decrease of maximum data rate should be included and would be very suitably placed below eq(12).

Comment 6). The information on the stray capacitance values is valuable and would be interesting to the reader. I propose to place it after "Ignoring the stray capacitance of coils and semiconductors" writing "(amounting to 13 and 22 pf, respectively, for the laboratory prototype)".